# Data Collection for the Fourth Multicentre Confidential Enquiry into Perioperative Equine Fatalities (CEPEF4) Study: New Technology and Preliminary Results

**DOI:** 10.3390/ani11092549

**Published:** 2021-08-30

**Authors:** Miguel Gozalo-Marcilla, Regula Bettschart-Wolfensberger, Mark Johnston, Polly M. Taylor, Jose I. Redondo

**Affiliations:** 1Veterinary Clinical Sciences, The Royal (Dick) School of Veterinary Studies and The Roslin Institute, Easter Bush Campus, The University of Edinburgh, Edinburgh EH25 9RG, UK; 2Department of Clinical Diagnostics and Services, Vetsuisse Faculty, University of Zürich, 8057 Zürich, Switzerland; rbettschart@vetclinics.uzh.ch; 3Vetstream Ltd., Three Hills Farm, Bartlow, Cambridge CB21 EN, UK; mark.johnston@vetstream.com; 4Taylor Monroe, Little Downham, Cambridge CB6 2TY, UK; polly@taylormonroe.co.uk; 5Departamento de Medicina y Cirugía Animal, Facultad de Veterinaria, Universidad Cardenal Herrera-CEU, CEU Universities, 46115 Valencia, Spain; nacho@uchceu.es

**Keywords:** anaesthesia, CEPEF, data analysis, death, epidemiology, equine, horse, mortality, standing sedation

## Abstract

**Simple Summary:**

New technologies allow researchers to improve the methods for immediate, accurate data collection, cleaning and analysis, with minimal geographical limitations. Although much has improved in the field of equine anaesthesia in recent years, we are still far from reducing anaesthetic-related mortality in this species in comparison with small animal anaesthesia. The aim of this multicentre study was to probe the usefulness of an internet-based method that utilised an electronic questionnaire and statistical software to show the data and report outcomes from horses undergoing general anaesthesia and certain procedures using standing sedation. Within six months, 8656 cases from 69 centres were collected: 6701 procedures under general anaesthesia and 1955 under standing sedation. The results demonstrated (i) the utility of the method and (ii) that some horses died unexpectedly when undergoing not only general anaesthesia, but also standing sedation. Finally, (iii) we present some descriptive data that outline the current anaesthesia practice compared with the previous CEPEF2. We concluded that our internet-based method is suitable for this type of study. New techniques may reduce the mortality rate. However, the results presented here should be interpreted cautiously as these are only preliminary data with lower numbers than CEPEF2.

**Abstract:**

It is almost 20 years since the largest observational, multicentre study evaluating the risks of mortality associated with general anaesthesia in horses. We proposed an internet-based method to collect data (cleaned and analysed with R) in a multicentre, cohort, observational, analytical, longitudinal and prospective study to evaluate peri-operative equine mortality. The objective was to report the usefulness of the method, illustrated with the preliminary data, including outcomes for horses seven days after undergoing general anaesthesia and certain procedures using standing sedation. Within six months, data from 6701 procedures under general anaesthesia and 1955 standing sedations from 69 centres were collected. The results showed (i) the utility of the method; also, that (ii) the overall mortality rate for general anaesthesia within the seven-day outcome period was 1.0%. In horses undergoing procedures other than exploratory laparotomy for colic (*“noncolics”*), the rate was lower, 0.6%, and in *“colics”* it was higher, at 3.4%. For standing sedations, the overall mortality rate was 0.2%. Finally, (iii) we present some descriptive data that demonstrate new developments since the previous CEPEF2. In conclusion, horses clearly still die unexpectedly when undergoing procedures under general anaesthesia or standing sedation. Our method is suitable for case collection for future studies.

## 1. Introduction

Internet-based research has become commonplace during the last 10–20 years and makes online data collection possible from a large pool of participants with few geographical limitations. Online data collection is quick, cheap and increases the accuracy and efficacy of data entry [1]. Moreover, data can be analysed interactively with the ability to follow-up with participants [1,2,3]. Apart from understanding how to use new technology and the expertise required to match the study design with data collection, adequate statistical software for data analysis and cleaning is mandatory [4,5]. Immediate technical support and fluent communication between the administrators and participants is also crucial. 

The high risk of mortality associated with general anaesthesia in horses remains one of the biggest concerns for equine practitioners and veterinary anaesthetists. Many studies report anaesthesia-related mortality risks. Most of these studies are retrospective [6,7,8,9,10,11,12,13,14] but a few prospective single-centre [15] and some prospective, multi-centre [16,17] investigations have been carried out. To date, the Confidential Enquiry into Perioperative Equine Fatalities 2 (CEPEF2) published in 2002 remains the largest observational, multicentre study with 41,824 cases collected from 62 clinics over a period of 6 years [18]. The overall death rate up to seven days was 1.9%, 0.9% in noncolics and 7.8% in colics [18]. Although much has changed since then, we are still far from reducing these numbers [19], and the need for an update on the CEPEF data was proclaimed eight years ago [20]. Avoiding general anaesthesia by undertaking some procedures in standing horses may reduce the mortality, but there are no data as yet to support this hypothesis. 

The first objective of this report was to describe the usefulness of an internet-based method for data collection and a strategy for data cleaning for the CEPEF4 study, whose final aim is to identify the risk factors associated with equine anaesthesia and standing sedation. Second, to report the outcomes/fatalities within a period of seven days of equidae undergoing (i) general anaesthesia and (ii) certain procedures performed under standing sedation during the first six months of CEPEF4 and to describe the key preliminary findings that will be analysed in detail in the future. Our first hypothesis was that the proposed method would be a fast, reliable tool to collect, clean and analyse data. Second, that mortality was lower than reported by Johnston et al. almost 20 years ago [18] and that standing sedation procedures are not exempt from the risk of death, although it is reduced. Finally, we hypothesised that there would be trends towards new practices in anaesthesia and analgesia compared with the previous CEPEF2.

## 2. Materials and Methods

The study design was multicentre, cohort, observational, analytical, longitudinal and prospective. For this phase of preliminary results, the six-month data collection period was from 1 November 2020 until 30 April 2021.

Inclusion criteria comprised cases of horses, donkeys and mules of all ages from clinics all over the world, recruited specifically for this project. The equidae involved were all cases from each clinic undergoing (i) general anaesthesia whatever the reason and (ii) standing sedation for surgery or advanced diagnostic imaging (magnetic resonance imaging (MRI), computed tomography (CT) or scintigraphy) requiring a continuous rate infusion (CRI) or at least one extra top-up apart from the initial sedation bolus. Once recording started in any one clinic, all cases had to be included. However, to allow for holidays and staff absence, all cases within a specified period could be omitted; recording and sending all cases started again after the specified period.

Exclusion criteria were cases outside the agreed recording periods, cases from centres that did not follow the communication process and cases from centres that did not send all the cases from a recording period. General anaesthetics for terminal procedures were excluded, as well as standing sedations without top-ups or CRIs, and standing sedations for reasons other than surgery or advanced diagnostic imaging, such as cast changes or sinuscopies.

The steps followed in this study are detailed below:(1)Preparation of the questionnaire

A user-friendly, online questionnaire used in small animals [21] was adapted for this equine study using the feedback received from a group of researchers and clinicians with special interest in equine anaesthesia and analgesia [22]. It was designed to be used to compile information for both general anaesthesia and standing sedation. 

Briefly, the questionnaire collected data about the centre, the level of training of the responsible anaesthetist, the patient, the procedure, the anaesthetic and analgesic protocols and other details about management of the anaesthesia or sedation. Information about the anaesthetic induction and recovery phases was included, with the potential to indicate any intraoperative complications and details of the postoperative period for up to seven days. The questionnaire and the instructions to complete it are available on a website created for support and promotion at https://cepef4.wordpress.com (accessed on 6 June 2021), and are provided here as Appendix A.

(2)Definition of perioperative equine fatalities

The same classification of fatalities was followed as used in previous CEPEF studies [16,18]. After induction of general anaesthesia or the first sedation bolus for standing sedation, the outcome was recorded at day seven for each case as (i) *alive (or discharged)*, (ii) *put to sleep (PTS)/euthanised* or (iii) *dead*. The time of PTS or dead was recorded as premedication, induction, maintenance, recovery or the day up to day seven. Each centre was encouraged to use an outcome logbook to communicate further which colic or noncolic cases were alive, PTS or dead during this seven-day period.

When a horse was euthanized due to an inoperable lesion found at surgery, due to pre-existing disease or financial constraints, the outcome was classified as *PTS*. When a horse died unexpectedly or was euthanised due to a perioperative complication, such as a fracture in recovery, the outcome was classified as *death*. Other examples of death include intraoperative cardiac arrest, spinal cord malacia requiring euthanasia during recovery or post-operative myopathy requiring euthanasia on humane grounds up to seven days after surgery. Those classified as noncolic deaths were required to complete a more detailed online survey at https://edinburgh.onlinesurveys.ac.uk/equine-ga-mortality-form (accessed on 6 June 2021).

Classification of outcomes was performed by M.G.-M. and J.I.R., and later confirmed by R.B.-W., M.J. and P.M.T. Ultimately, before final data analysis, the outcomes were confirmed with the contact person from each centre, referred to as the ambassador (see (10) in Methods).

(3)Ethics statement

The study was approved by the international Ethical Review Committee of the Association of Veterinary Anaesthetists (AVA), under protocol 2020-009 on 4 September 2020. Full details of the application form were provided to any centre that also required Local Ethical Committee Approval from their institution.

(4)Recruitment of collaborating centres

Recruitment included an abstract presented at the AVA Spring meeting in Dublin 2020 [23], and Correspondence to the Editor published in peer reviewed journals targeting equine practitioners [24] and veterinary anaesthetists [22], respectively. Finally, we also used our professional network and contacted several centres personally. 

(5)The Ambassador figure and the Agreement Form

Ambassadors for each centre volunteered or were recruited to be the contact person who also took responsibility for the centre’s data collection.

Before sending the data, the ambassador signed an agreement form (https://cepef4.wordpress.com/forms-instructions-and-help/ (accessed on 6 June 2021)) ensuring to cooperate/supervise selflessly, providing good quality data including all the cases within a period of time to avoid bias. The CEPEF team certified the anonymous handling of the data.

(6)Anonymity and confidentiality of each patient, owner and centre

As in previous CEPEF studies, the anonymity and confidentiality of the patients, owners and centres was ensured. Identification of each patient was provided by a number given by the centre for further communication with the CEPEF team. These numbers were used only for communication with each centre. Each case entered in the database received a unique CEPEF number for data handling and analysis. Moreover, each centre had a code known only by M.G.-M. and J.I.R. The remaining authors, R.B.-W., J.I.R. and P.M.T., were not aware of centre coding for further evaluation of the data. 

(7)The communication process (recruitment and follow-up meetings)

We instituted a method of interactive communication with the ambassador of each centre always performed by the same investigator (M.G.-M.) in order to standardize the information and, therefore, reduce data inconsistencies. For each centre, the project was first presented to the ambassador, and ideally, also to the whole team involved in data collection, via an online meeting (Teams^®^ (Microsoft Corporation, Redmond, WA, USA), Zoom^®^ (Zoom Video Communications Inc., San Jose, CA, USA), Skype^®^ (Microsoft Corporation, Redmond, WA, USA), Meet^®^ (Google Inc., Mountain View, CA, USA)) of about one hour’s duration. This included a standard presentation of the project, introduction to the study website and explained the questionnaire and its technicalities. At least a second, and ideally a third, follow-up meeting was subsequently organised to ensure complete familiarity with the system. 

(8)Data collection and storage

The electronic questionnaire was used to collect the data (Appendix A). A .pdf file was programmed using Adobe Acrobat Pro^®^ (Adobe Inc., San Jose, CA, USA). This .pdf file can be completed using the free app Adobe Acrobat Reader DC (Adobe Inc., San José, CA, USA), available for the different operating systems (Android^®^ (Google Inc., Mountain View, CA, USA), IOS^®^ (Apple, Cupertino, CA, USA), Windows^®^ (Microsoft Corporation, Redmond, WA, USA) and macOS^®^ (Apple, Cupertino, CA, USA)). The questionnaire can be completed on any internet-connected device, including mobile phones, tablets, laptops or computers. Once completed, the questionnaire is sent by e-mail to a specific e-mail account in which all the questionnaires are stored. The collected metadata is converted to a .csv file that is added to the main database as a new case. 

(9)Data cleaning and statistics

Data processing and statistical analyses were performed using R 4.1.0 [25]. Data quality was improved using an initial data cleaning phase performed with specific scripts searching for inconsistencies. R scripts were introduced to detect duplicated cases or blank fields as non-available data (-NA-) in the studied variables. Other R scripts detected more sophisticated inconsistencies using Boolean algebra, for instance, in general anaesthetics coded as inhalation anaesthesia only (INH) but where an alpha_2_-agonist CRI was ticked, or when an inhalant agent was ticked for a standing sedation procedure. In addition, further R scripts were used to detect uncommon anaesthetic practices, such as induction with an inhalant agent or when premedication was not administered. The full list of the R scripts used in this study can be found in the Appendix A.

The ambassador of any centre where data inconsistencies were detected was contacted and sent an Excel^®^ (Microsoft Corporation, Redmond, WA, USA) file by e-mail to confirm whether the inconsistencies were real and to request revision if so. These detected inconsistencies were corrected manually within the database, only with the strict agreement of the ambassador.

Once data were cleaned, a descriptive analysis was performed. Tables and figures were created using the following R packages: table1 (v1.4.1) [26], ggplot2 (v3.3.3) [27] and rnaturalearth (v0.1.0) [28]. Variables that followed a normal distribution are shown as mean ± standard deviation, whereas those non-normally distributed are shown as median [range].

(10)Final meeting with each ambassador to double-check data and outcomes

The individual database from each centre was sent to the ambassador of the centre before the final meeting. This included all the cases sent up until 30 April 2021. A personalized report for the centre was included for its approval. Data were always double-checked by the principal investigator (M.G.-M.) and the ambassador for each centre during a scheduled online meeting of about 30 to 90 min, depending on the volume of the data and potential inconsistencies. The ambassadors were asked for feedback and to grade the project and the communication process between 0 (worse) to 10 (excellent). These meetings occurred between Monday 17 May and Thursday 3 June 2021.

Figure 1 shows the flow diagram of the process.

(11)Reporting the results

The STROBE-Vet guidelines, an extension of the STROBE (Strengthening the Reporting of Observational Studies in Epidemiology) statement [29], as recommended for reporting of observational studies in veterinary medicine (https://strobevet-statement.org, accessed on 6 June 2021), (Appendix A) were followed in order to maximise reporting quality.

## 3. Results

During the specified first six months, 69 centres from 20 countries in four continents collaborated to collect data (Figure 2, Table 1). As a result of continuous recruitment, each centre had a different starting date for data collection.

Figure 3 shows the flow diagram describing the cases included or excluded from the current analysis. The data cleaning process detected 193 empty data fields and 1310 inconsistencies. After a discussion with the ambassador of each centre, 50 inconsistencies were redeemed, but were noted as unusual practices. Sixty two empty data fields and 1260 inconsistencies were corrected. For standing sedations, 131 durations were not found in the records of the centres and were coded as missing data. 

After data cleaning, 8752 cases (8656 horses, 92 donkeys and 4 mules) were confirmed. The median and range of cases received per day were 48 [1–146]. These preliminary results contain horse information only as the sample size for donkeys and mules is too small at this stage.

Of the 8656 horse cases, 6701 were general anaesthetics and 1955 were standing sedations. Of these, only 39 cases were procedures in the field, 31 total intravenous anaesthetics and 8 standing sedations. The demographic data for general anaesthetics and standing sedation are shown in Table 2.

### 3.1. Perioperative Equine Fatalitiess

Sixty six of the 6701 horses that underwent general anaesthesia were classified as having died (1.0%) (confidence interval 95% (CI 95%): 0.76–1.25%). Of these, 35 (out of 5784) were classified as noncolic deaths (0.6%) (CI 95%: 0.42–0.84%) and 31 (out of 917) as colic deaths (3.4%) (CI 95%: 2.3–4.8%). Four of the 1955 horses undergoing standing sedation died (0.2%); all were noncolic surgeries (0.2%) (CI 95%: 0.06–0.52%). 

Of the 6701 horses that underwent general anaesthesia, 329 were classified as PTS (4.9%) (CI 95%: 4.4–5.5%): 76 of 5784 as noncolic PTS (1.3%) (CI 95%: 0.88–1.45%) and 253 of 917 as colic PTS (27.6%) (CI 95%: 25.0–31.0%). From the 1955 horses undergoing standing sedation, 13 were PTS (0.7%) (CI 95%: 0.35–1.13%): 11 noncolic surgeries out of 1949 (0.6%) (CI 95%: 0.28–1.01%) and 2 out of 6 colic surgeries (33.3%) (CI 95%: 4.3–77.7%).

The outcomes at seven days post-anaesthesia or standing sedation are shown in Table 3. Details of the deaths classified as noncolics are presented in Table 4. Table 5 indicates the time of death or PTS of all the cases whether general anaesthesia or standing sedation was used. 

### 3.2. General Anaesthesia

The centres sent a median of 100% of cases (100 [95–100]%). Of the 69 centres, 63 confirmed that 100% of the cases were sent, three sent more than 98% and three sent more than 95%.

Table 6 gives the details of the individual drugs used at each phase of general anaesthesia. The protocols for the general anaesthetics are described in Table 7, Table 8, Table 9, Table 10 and Table 11. Information about the method of induction and recovery from general anaesthesia is shown in Table 12 and Table 13, respectively. Figure 4 describes the monitoring of the horses under general anaesthesia. 

### 3.3. Standing Sedation

From the 69 collaborating centres, 57 sent standing sedation cases that fulfilled our inclusion criteria (median 99 [10–100]%). Forty-three centres sent more than 90% of their cases, four sent between 80 and 90%, five sent between 50 and 80% and five sent between 1 and 49%. 

The drugs and protocols for standing sedations are shown in Table 14 and Table 15, respectively. Table 16 reports the different CRIs used to maintain standing sedation. Figure 5 describes the monitoring of the horses under sedation.

### 3.4. Results of Survey for Feedback in Final Meeting

The ambassadors of the collaborating centres gave a final mark for the project and the communication process of median 10 [7–10] (0—worse to 10—excellent).

## 4. Discussion

The results presented here demonstrate the utility of the electronic questionnaire and the internet-based method for data collection, with interactive data handling and cleaning for the performance of the CEPEF4 study. The data collected from the 1 November 2020 until the 30 April 2021 show that overall, 1% of horses undergoing general anaesthesia still die within the seven-day outcome period. Even with this preliminary evaluation of a small number of horses, it appears that a fatal outcome in both noncolics (0.6%) and colics (3.4%) was less frequent than that reported by Johnston et al. almost 20 years ago [18]. As anticipated, and despite the higher risk of bias collecting standing sedation data, the risk of death with standing sedation (0.2%) appears lower than with general anaesthesia. However, even with a lower risk, some noncolic horses died unexpectedly when undergoing standing sedation within the seven-day outcome period. The data also demonstrate certain changes in the routine anaesthetic practice and protocols that have developed since CEPEF2.

In light of our findings, and focussing on our main objectives, the discussion is structured in the following four parts: (1) the proposed method for data collection with interactive data handling and cleaning, (2) the outcomes of horses that underwent general anaesthesia and standing sedation, (3) the preliminary data from horses undergoing general anaesthesia and (4) the preliminary results of standing sedation in horses. 

(1)The proposed method for data collection with interactive data handling and cleaning

Our first hypothesis was confirmed as the proposed internet-based method proved to be a reliable, easy, quick and cheap means of collecting data, with minimal geographical limitations. This preliminary phase of CEPEF4 was initiated and executed during the COVID-19 pandemic but our methodology allowed us to communicate with the collaborating centres and to collect and handle data arising from many locations worldwide. The feedback from the ambassadors of each centre indicated that the communication process (recruitment, follow-up and final meetings) allowed fluent communication that undoubtedly contributed to the collection of good quality data [2]. Once collected, our R scripts detected many inconsistences that were reduced almost to zero prior to final submission. The three steps followed for this data cleaning proved effective: first, data screening to detect missing or excess data, outliers, inconsistencies or strange patterns. Second, diagnosis to detect missing data, true extremes or true errors. Finally, data editing or treatment to correct, delete or leave unchanged the detected inconsistencies [5] as agreed with each centre’s ambassadors. Even with this large dataset, we achieved a reliable method with a clear strategy for communication and data collection/cleaning. This approach will help to minimize inconsistencies that might lead to data misinterpretation for the current CEPEF4 study. 

For the first six months, we compiled 8656 horse cases: 6701 general anaesthetics and 1955 standing sedations. CEPEF1 collected 6255 general anaesthetics in a period of two years (February 1991–March 1993), using paper copies and communication by post and telephone [16]. This reflects the importance of new technologies that allow interactive communication, data collection and analysis. Furthermore, the rapid availability of a community of researchers and equine clinicians familiar with the CEPEF studies made this work possible in such a short period of time. The impact of the CEPEF2 study, cited by 448 other publications (according to Google Scholar on the 29 August 2021), reflects the importance of an update on these data [20]. 

With the first hypothesis confirmed, our final goal is to use this methodology to collect approximately 45,000 cases of general anaesthetics for CEPEF4, to increase the statistical power and to compare the results with those of 20 years ago [18]. In order to ensure a robust comparison, we followed the same approach as in CEPEF2. Both M.J. and P.M.T., authors of the previous series, carefully supervised the study design and the subsequent implementation of data collection and cleaning. On the basis of the cumulative cases collected per week to date, the proposed cases numbers should be reached in approximately two years.

(2)Outcome of horses under general anaesthesia and standing sedation

As reported in CEPEF2 [18], the overall equine anaesthetic mortality rate is still higher than in other veterinary species such as dogs and cats [30]. However, these preliminary data from a small population do suggest that the current rate is lower than 20 years ago. 

As stated in 2016 by Dugdale and Taylor [19], *“we still lose horses after anaesthesia to a range of catastrophes that would not occur if the horse were not anaesthetized”*. Our preliminary results confirm this statement. We recorded 39 horses that were classified as noncolic deaths: 34 general anaesthetics, four standing sedations and one that started as standing sedation and later changed to general anaesthesia. 

Focusing on the noncolics undergoing general anaesthesia, with the exception of 17 cases classified as moderate- or high-risk patients (ASA III, IV or V), the remaining 18 were healthy (four ASA I, 14 ASA II). This indicates that noncolic horses undergoing general anaesthesia still die unexpectedly; 0.6% at this stage, *versus* the 0.9% of CEPEF2. Further recruitment of cases to achieve our final CEPEF4 goal should help to clarify why this occurs and hopefully suggest how these numbers could be reduced. 

With regard to colics undergoing general anaesthesia, our preliminary data recorded 3.4% that died, compared to the 7.8% (457 colic deaths out of 5846) in CEPEF2. This relative reduction may indicate an improvement in the peri-operative management of horses with colic. However, the number of colic PTSs remain extremely high, 27.6% in this data *versus* the 25.2% of CEPEF2. 

It is generally assumed that standing procedures are safer than general anaesthesia in horses. However, this is the first study to report mortality rates associated with standing sedation in a multicentre, prospective, cohort study. The four standing sedations (one ASA I, one ASA II and two ASA III) classified as noncolic deaths (0.2%) provide evidence that standing sedation procedures still carry a risk, albeit less than general anaesthesia.

(3)Preliminary results for general anaesthetics in horses

One third of the horses were premedicated with a combination of an alpha_2_-agonist and a partial/agonist-antagonist opioid, mainly butorphanol. When acepromazine was also given, the percentage increased to 48%. Alpha_2_-agonists with a pure opioid, with or without acepromazine, was used in about 26% of the cases. Only 13% of the cases received an alpha_2_-agonist alone. This is in contrast with the CEPEF2 data, in which 43% of the horses received an alpha_2_-agonist alone, 29% combined acepromazine with an alpha_2_-agonist and no more than 8% included opioids such us methadone or butorphanol [18]. Our data show the tendency for premedication with drug combinations rather than a single agent. Drug combinations calm the horse, enhance sedation and provide analgesia by employing low doses of each drug, which reduces their potential side effects [31,32,33]. Compared with around 16% in CEPEF2 [18], premedication with acepromazine alone has not been reported in the current study.

Ketamine was the most commonly used intravenous agent for the induction of general anaesthesia, usually combined with a benzodiazepine as a central muscle relaxant (88%). Diazepam was frequently used, and increasingly midazolam, which was not reported at all in CEPEF2. Midazolam has recently gained Market Authorisation for use in horses in Europe, which would support its increased use. Thiopental and guaiphenesin, an intravenous anaesthetic and a central muscle relaxant, respectively, were rarely used. This combination was historically the evolution from chloral hydrate and pentobarbital [34]; however, the often prolonged and violent recovery from barbiturate anaesthesia provoked the transition towards ketamine [35]. A few techniques were popular in some individual clinics, although the overall numbers were not high. For example, the induction and brief maintenance of general anaesthesia in foals with propofol, sometimes combined with ketamine [36]. 

Isoflurane was the most common inhalation agent used for the maintenance of general anaesthesia (88%), followed by sevoflurane (9%) and desflurane (3%). None of the anaesthetics reported in these preliminary results used halothane in contrast with the previous CEPEF studies [16,18]. In 2004, halothane was considered an acceptable anaesthetic for the maintenance of anaesthesia in horses [37]; however, it is no longer manufactured and is now rarely used. Our data also showed the current tendency to use an inhalant agent in combination with an intravenous CRI, the so-called partial intravenous anaesthesia (PIVA) [38,39,40]. Out of all the 6000 inhalant-based general anaesthetics, 3718 were PIVA (62%) *versus* 2282 (38%) that were pure inhalation anaesthesia. Since the publication of CEPEF2 nearly 20 years ago, there have been numerous reports of PIVA techniques using alpha_2_-adrenergic agonists [41,42], lidocaine [43], ketamine [44] and opioids [45,46]. 

Only 701 of the total 6701 were general anaesthetics where total intravenous anaesthesia (TIVA) was used, 31 under field conditions. Very short procedures were carried out in the anaesthetic recovery box or in the field using repeated boluses of a range of drugs and combinations. Ketamine, thiopental, alpha_2_-agonists and benzodiazepines were all used. Less than 300 of the 701 TIVA cases were maintained with various combinations of the so-called *“triple drip”*, usually combining ketamine with an alpha_2_-agonist and guaiphenesin [40]. The maintenance of anaesthesia for short procedures was also carried out with ketamine alone or occasionally with thiopental.

Our data show that it is now common practice (63%) to administer a small dose of an alpha_2_-agonist after the end of general anaesthesia before the recovery phase, either when the patient is still on the surgical table and before transport to the recovery box, or once in the recovery box. Santos et al. (2003) [47] first demonstrated its benefits, later confirmed by others [48,49]. 

This study collected relevant information about the monitoring undertaken during general anaesthesia. Although it has been recently suggested that improvements in monitoring have reduced the risk of anaesthetic mortality [19], no data from a prospective, multicentre study support this statement. Our data indicated that around 90% of the horses undergoing general anaesthesia (INH, PIVA or TIVA) had an electrocardiogram and a pulse oximeter. Arterial blood pressure (invasively) and the end-tidal carbon dioxide (EtCO_2_) concentration were measured in more than 75%. Inspired oxygen and an end tidal volatile agent were measured in fewer cases, but still more than 50%. Arterial blood gas analyses (70%) and lactate (43%) were performed most often in horses with colic. In general, body temperature was not frequently monitored (33%), and the use of spirometry in horses was minimal. Non-invasive blood pressure was sometimes used in foals, when an arterial line had not been placed, or even with invasive blood pressure. These data are similar to those reported via the online questionnaire in 2015 by Wohlfender et al. [50].

The data show the current protocols for the induction of anaesthesia were usually assisted, with either personnel (61%) or using a gate (25%). For recovery, 51% were free, 41% were assisted with ropes and 8% were manually assisted, mostly foals. These data reflect the ongoing controversy about which is the best method of recovery from general anaesthesia. Head and tail rope systems do not completely prevent fractures during recovery [51,52,53], but some cases may benefit from this technique. Slings were used in some cases but again did not ensure a good recovery: one young horse became excited with sling assistance and recovered badly after an elbow fracture repair; this was one of the unexpected noncolic deaths. 

(4)Preliminary results of standing sedations in horses

The data presented here are the first reported from a prospective, multicentre, cohort study of standing sedation, but the numbers are small and cautious interpretation is still in order.

Whereas the data collected from general anaesthetics can be considered as strong and reliable, only 43 of the 57 centres sent more than 90% of their standing sedation cases, which could lead to biased data. The main challenge for centres with a high case load was to collect every single case. 

Alpha_2_-agonists were used in all except three of the premedication combinations. Acepromazine was not given alone but, combined with at least an alpha_2_-agonist, it was given to 47% of the cases. Opioids, mostly butorphanol, morphine and methadone were used in 85% of the cases. Boluses of detomidine, butorphanol and romifidine are the most commonly used for the maintenance of standing sedations. For CRIs, detomidine was the most commonly used, followed by far, by butorphanol. Monitoring in standing sedation procedures was minimal. 

Many of the procedures previously performed only in anaesthetised horses are now carried out using standing sedation. Although the inclusion criteria were more restrictive and not all the centres were able to supply these procedures, 23% of the cases in this study were standing sedations. Further study of these data is required. Creating a subgroup of collaborating centres may ensure more reliable data.

(5)Limitations

Our study is not free of limitations. First, the electronic questionnaire. Its use required a learning curve and sometimes raised technological difficulties, all solved with online support. The potential for creating a specific app is under consideration; however, the adaptation to different software can be challenging. Second, the current version of the questionnaire could be improved. For instance, the field *“castrated”* and *“pregnant”* is not mandatory in the current version and could lead to misinterpretation. Drugs commonly used, such as mepivacaine, were not included (but could be added manually). An updated version will be created in response to the feedback from the collaborating centres. Third, there is an inherent bias as some areas of the world and types of practice are underrepresented. Further strategies are to be implemented in this respect, although language barriers should not be underestimated. Finally, the amount of information collected by the questionnaire is enormous and cannot be covered in a single scientific paper. However, and as stated in the agreement form, *“sub-studies can be proposed, as long as this does not involve duplicate use of the CEPEF4 data”*. This may allow our community to benefit from our method for collecting multicentric data for purposes other than mortality up to seven days.

## 5. Conclusions

We have designed a reliable method, with a clear strategy of communication and data collection/cleaning that can be used to collect cases for CEPEF4. This approach will help to minimize inconsistencies that may lead to data misinterpretation in future CEPEF studies. This preliminary report shows that horses still die unexpectedly during and within the seven-day postoperative period of general anaesthesia and standing sedation. Our results also show that current practice in anaesthesia has changed over the last 20 years.

## Figures and Tables

**Figure 1 animals-11-02549-f001:**
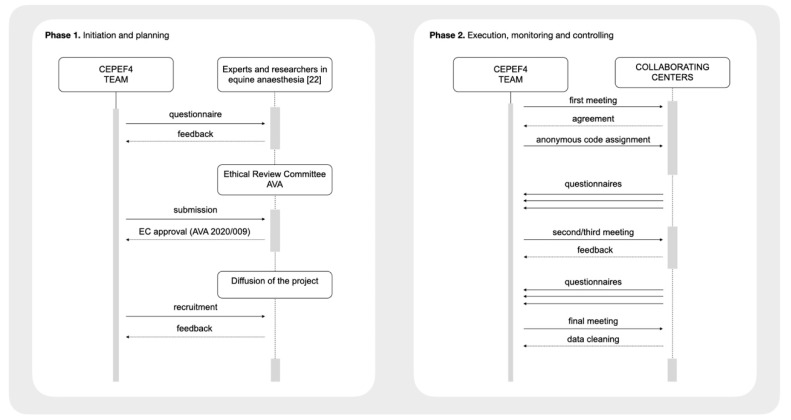
Flow diagram of the process.

**Figure 2 animals-11-02549-f002:**
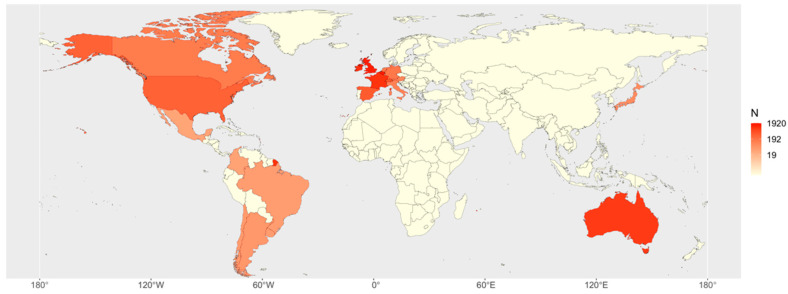
Heat map of the distribution of the cases by country.

**Figure 3 animals-11-02549-f003:**
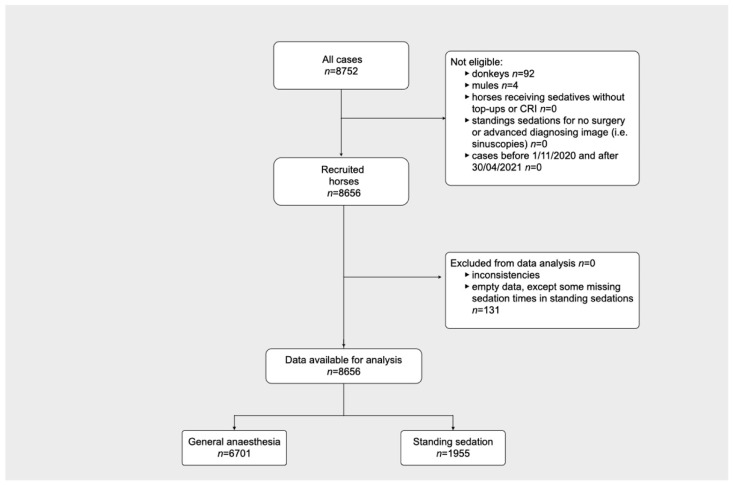
Flow diagram describing the cases included or excluded in the study.

**Figure 4 animals-11-02549-f004:**
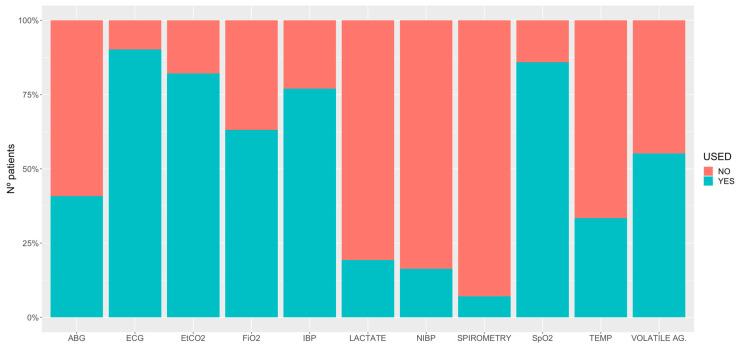
Percentage of monitoring used in 6701 horses undergoing general anaesthesia. ABG, arterial blood gases; ECG, electrocardiogram; E_t_CO_2_, end-tidal carbon dioxide; FiO_2_, inspiratory fraction of oxygen; IBP, invasive blood pressure; NIBP, non-invasive blood pressure; SpO_2_, partial saturation of haemoglobin with oxygen by pulse-oximetry; TEMP, temperature; VOLATILE AG., volatile agents measured for inspired fraction and end-tidal concentration.

**Figure 5 animals-11-02549-f005:**
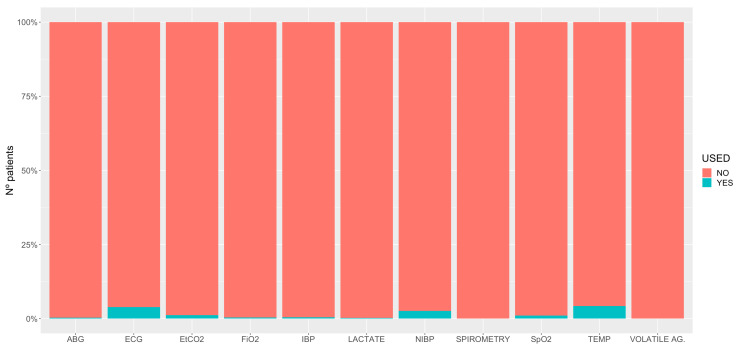
Percentage of monitoring used in 1955 horses undergoing standing sedation. ABG, arterial blood gases; ECG, electrocardiogram; E_t_CO_2_, end-tidal carbon dioxide; FiO_2_, inspiratory fraction of oxygen; IBP, invasive blood pressure; NIBP, non-invasive blood pressure; SpO_2_, partial saturation of haemoglobin with oxygen by pulse-oximetry; TEMP, temperature; VOLATILE AG., volatile agents measured for inspired fraction and end-tidal concentration.

**Table 1 animals-11-02549-t001:** Countries of origin of the 69 collaborating centres classified per type of hospital/clinic and cases and percentage of cases registered for each country.

Country	University Hospitals	Referral Centres	Ambulatory Clinicians	Total Centres	Cases ^1^	% Cases
Belgium	2	2	0	4	1906	21.78
United Kingdom	3	10	0	13	1670	19.08
Republic of Ireland	1	2	0	3	1068	12.20
France	0	6	0	6	1025	11.71
Australia	0	2	0	2	898	10.26
Switzerland	2	2	0	4	444	5.07
Spain	6	2	0	8	399	4.56
United States of America	3	0	0	3	327	3.74
Germany	1	0	0	1	162	1.85
The Netherlands	1	1	0	2	150	1.71
Canada	2	0	0	2	142	1.62
Italy	2	0	0	2	120	1.37
Austria	1	0	0	1	97	1.11
Japan	0	1	0	1	82	0.94
Argentina	1	1	0	2	55	0.63
Chile	1	0	0	1	48	0.55
Uruguay	1	1	1	3	46	0.53
Brazil	4	0	0	4	44	0.50
Colombia	3	0	1	4	35	0.40
Mexico	0	2	1	3	34	0.39
Total	34	32	3	69	8752	100

^1^ The total number of cases includes horses, donkeys and mules.

**Table 2 animals-11-02549-t002:** Demographic data by general anaesthesia or standing sedation and overall, in 8656 horses.

Variable	Categories	General Anaesthesia (*n* = 6701)	Standing Sedations (*n* = 1955)	Overall (*n* = 8656)
Sex	Female	2699	817	3516
	Male	4002	1138	5140
Age	Neonate (≤1 month)	229	30	259
	Foal (1–12 months)	1407	27	1434
	Young (1–5 years)	2314	328	2642
	Adult (5–14 years)	2029	1131	3160
	Geriatric (>14 years)	722	439	1161
BCS	Normal	5766	1585	7351
	Thin	415	129	544
	Fat	520	241	761
ASA	I	3007	1200	4207
	II	2354	666	3020
	III	546	79	625
	IV	470	9	479
	V	324	1	325
Reason ^1^	Abdominal	218	61	279
	Colic	917	6	923
	Diagnostic	479	959	1438
	ENT	377	138	515
	Fracture	166	28	192
	Orthopaedics	3346	138	3484
	Urogenital	820	103	923
	Miscellaneous	898	573	1471
Colic/noncolic	Noncolic surgery	5784	1949	7733
	Colic surgery	917	6	923
Type of anaesthesia	Standing sedation	0	1955	1955
	Inhalatory	2282	0	2282
	PIVA	3718	0	3718
	TIVA	701	0	701
Duration *	<1 h	2317	594	2911
	1–2 h	2815	744	3559
	2–3 h	1123	333	1456
	>3 h	446	153	599
Use of locoregional techniques	No	5602	1114	6716
	Yes	1099	841	1940
Use of mechanical ventilation	No	1870	1955	3825
	Yes	4831	0	4831
Timetable	Normal	5866	1918	7784
	Out of hours	835	37	872
Scheduling	Scheduled	5120	1817	6937
	Non scheduled	380	80	460
	Urgent	1201	58	1259

^1^ Horses could be anaesthetized/sedated for more than a reason. ASA, American Society of Anaesthesiologists; BCS, body condition score; ENT, ear–nose–throat; PIVA, partial intravenous anaesthesia; TIVA, total intravenous anaesthesia. * 131 durations in standing sedations were coded as missing data.

**Table 3 animals-11-02549-t003:** Data from 6701 equine anaesthetics and 1955 equine standing sedations recorded: reason for anaesthesia/standing sedation and outcome at 7 days post-anaesthesia. Horses could be anesthetized or sedated for more than a reason.

	Alive	Dead	% Deaths	PTS	% PTS	Total
**General anaesthesia**						
Noncolic surgery	5673	35	0.6%	76	1.3%	5784
Abdominal	198	3	1.4%	17	7.8%	218
Diagnostic	440	3	0.6%	36	7.5%	479
ENT	373	4	1.1%	0	0.0%	377
Fracture	154	7	4.2%	4	2.4%	166
Miscellaneous	882	5	0.6%	11	1.2%	898
Orthopaedics	3306	13	0.4%	27	0.8%	3346
Urogenital	803	7	0.9%	10	1.2%	820
Colic surgery	633	31	3.4%	253	27.6%	917
Overall	6306	66	1.0%	329	4.9%	6701
**Standing sedation**						
Noncolic surgery	1934	4	0.2%	11	0.6%	1949
Abdominal	59	1	1.6%	1	1.6%	61
Diagnostic	950	0	0.0%	9	0.9%	959
ENT	137	0	0.0%	1	0.7%	138
Fracture	27	1	3.6%	0	0.0%	28
Miscellaneous	570	1	0.2%	2	0.4%	573
Orthopaedics	138	0	0.0%	0	0.0%	138
Urogenital	100	1	1.0%	2	1.9%	103
Colic surgery	4	0	0.0%	2	33.3%	6
Overall	1938	4	0.2%	13	0.7%	1955

ENT, ear–nose–throat; PTS, “put to sleep” (euthanized) horses.

**Table 4 animals-11-02549-t004:** Noncolic deaths under general anaesthesia and standing sedations.

Pre-Existing Condition and Brief Noncolic Death Description	*n*	Phase	Protocol
**General anaesthesia**			
ASA I and II	18		
Fracture in recovery (2 ASA I, 3 ASA II)	5	Recovery	2 PIVA, 2 INH, 1 TIVA
Post-operative colic (5 ASA II)	5	Days 3, 4, 5, 5 and 5	3 PIVA, 2 INH
Post-operative myelomalacia (2 ASA II)	2	Days 2 and 4	2 PIVA
Cardiac arrest (ASA I)	1	Maintenance	INH
Upper airway obstruction in recovery (ASA II)	1	Recovery	PIVA
Fracture C2 (ASA II)	1	Recovery	PIVA
Sudden collapse after standing (ASA I)	1	Recovery	TIVA
Presumed spinal cord malacia (ASA II)	1	Recovery	PIVA
Small colon prolapse (ASA II)	1	Day 1	PIVA
ASA III, IV and V	17		
Re-fractures in recovery (3 ASA IV)	3	Recovery	3 PIVA
Recumbent on arrival. Unable to stand after anaesthesia (2 ASA III)	2	Recovery	1 PIVA, 1 TIVA
Intraoperative bleeding without response to treatment (2 ASA III)	2	Maintenance	2 PIVA
Catastrophic recovery (<1 year with sling) (ASA III)	1	Recovery	PIVA
Post-operative colic (1 ASA III)	1	Day 3	INH
Re-fracture olecranon in the stable (ASA III)	1	Day 3	PIVA
Unknown, could not stand, no fracture diagnosed in X-rays (ASA III)	1	Recovery	INH
Severe chronic sinusitis. Sudden death. Necropsy: communication between sinus fistula and cranial cavity (ASA IV)	1	Recovery	TIVA
Foal with sepsis, diarrhoea (ASA IV)	1	Day 5	TIVA
Bladder rupture on caesarean section (ASA V)Fracture rib repair, hemothorax and hemopericardium (ASA V)	1	Maintenance	PIVA
	1	Maintenance	INH
Fracture/abductor tear after dystocia (ASA V)Intraoperative bleeding dystocia (ASA V)	1	Day 2	PIVA
	1	Day 1	INH
**Standing sedation**	4		
Post-operative colitis (ASA I, ASA II)	2	Days 1 and 6	CRI
Post-operative colic (ASA III)	1	Day 2	CRI
Re-fracture in stable (ASA III)	1	Day 1	CRI

ASA, American Society of Anaesthesiologists; CRI, continuous rate infusion; INH, inhalational anaesthesia only; PIVA, partial intravenous anaesthesia; TIVA, total intravenous anaesthesia.

**Table 5 animals-11-02549-t005:** Time of death of horses under general anaesthesia and standing sedation.

		PREM	IND	MAIN	REC	1D	2D	3D	4D	5D	6D	7D
Colic	Deaths	1	3	7	10	6	1	1	1	0	0	1
	PTS	0	0	186	12	14	15	9	5	9	0	5
Noncolic	Deaths	0	0	5	18	4	3	3	1	4	1	0
	PTS	0	0	38	3	15	12	8	4	3	1	3
Total	Deaths	1	3	12	28	10	4	4	2	4	1	1
	PTS	0	0	224	15	29	27	17	9	12	1	8

IND, induction; MAIN, maintenance; PREM, premedication; REC, recovery period; PTS, “put to sleep” (euthanized) horses. 1D–7D: days 1st to 7th.

**Table 6 animals-11-02549-t006:** List of the individual drugs used at each phase of general anaesthesia in a total of 6701 horses.

Drugs	PREM	IND	MAIN—(Bolus If Not Inhalant)	MAIN-CRI	POST
Acepromazine	2776	0	42	0	162
Xylazine	3231	0	327	724	2303
Detomidine	1569	0	44	274	340
Romifidine	1874	0	89	1132	1699
Medetomidine	236	0	81	471	287
Dexmedetomidine	5	0	65	296	52
Midazolam	87	2795	40	125	2
Diazepam	112	3293	17	0	2
Morphine	1270	0	239	7	222
Methadone	498	0	66	2	26
Pethidine	0	0	0	0	0
Fentanyl	6	0	4	3	0
Buprenorphine	2	0	1	0	5
Butorphanol	3320	0	261	238	94
Phenylbutazone	2647	0	64	0	657
Flunixin	2223	0	73	0	673
Meloxicam	413	0	5	0	143
Ketoprofen	93	0	1	0	6
Propofol	0	316	40	8	21
Alfaxalone	0	0	1	0	0
Ketamine	0	6633	1158	1090	37
Thiopental	0	95	234	0	23
TLT-ZLZ	0	41	1	0	0
GGE	0	84	44	442	0
Halothane	0	0	0	0	0
Isoflurane	0	1	5250	0	0
Sevoflurane	0	1	547	0	0
Desflurane	0	1	203	0	0
Lidocaine	17	0	0	1255	86
Dobutamine	0	0	0	4281	20
Phenylephrine	0	0	12	277	960

CRI, continuous rate infusion; GGE, guaiacol glyceryl ether; IND, induction; MAIN, maintenance; POST, immediate postoperative period; PREM, premedication; TLT-ZLZ, tiletamine-zolazepam.

**Table 7 animals-11-02549-t007:** Drugs and different drug combinations used for premedication before general anaesthesia in 6701 horses.

Drugs and Combinations	*n*	%
Alpha_2_ + Partial/Agonists-Antagonists Opioids	2214	33.0%
ACP + Alpha_2_ + Partial/Agonists-Antagonists Opioids	1026	15.3%
ACP + Alpha_2_ + Pure Opioids	1017	15.2%
Alpha_2_ alone	872	13.0%
ACP + Alpha_2_	721	10.8%
Alpha_2_ + Pure Opioids	704	10.5%
Partial/Agonist-Antagonist Opioid alone	62	0.9%
Pure Opioid alone	31	0.5%
Benzodiazepine alone	27	0.4%
ACP + Alpha_2_ + Pure Opioids + Partial/Agonists-Antagonists Opioids	11	0.2%
Alpha_2_ + Pure Opioids + Partial/Agonists-Antagonists Opioids	8	0.1%
ACP + Partial/Agonists-Antagonists Opioids	1	0.0%
None	7	0.1%

ACP, acepromazine; Alpha_2_, Alpha_2_-agonists. Benzodiazepines: 199 − 27 = 172 were used in different combinations but not alone.

**Table 8 animals-11-02549-t008:** Drugs used for induction of general anaesthesia in 6701 horses.

Induction Drugs	*n*	%
Ketamine + Benzodiazepine	5919	88.3%
Ketamine + Propofol	295	4.4%
Ketamine alone	247	3.7%
Ketamine + Thiopental	91	1.4%
Ketamine + GGE	80	1.2%
Tiletamine + Zolazepam	41	0.6%
Propofol	21	0.3%
Thiopental + GGE	3	0.1%
Ketamine + Inhalatory	1	0.0%
Inhalatory + GGE	1	0.0%
Thiopental alone	1	0.0%
Inhalatory	1	0.0%

GGE, guaiacol glyceryl ether. Benzodiazepines: 6088 − 5919 = 169 were used in different combinations but benzodiazepines + ketamine.

**Table 9 animals-11-02549-t009:** Drugs used for maintenance of general anaesthesia in 6701 horses.

Maintenance Drugs	*n*	%
Isoflurane	5250	78.3%
Sevoflurane	547	8.2%
Triple drip alone	277	4.1%
Desflurane	203	3.0%
Ketamine alone	87	1.3%
Thiopental alone	4	0.1%
Ketamine + Thiopental	3	0.0%
None	330	5.0%

**Table 10 animals-11-02549-t010:** Drugs used for continuous rate infusion during general anaesthesia with partial intravenous anaesthesia (PIVA) in 3718 horses.

CRI Drugs	*n*	%
Alpha_2_ alone	1780	47.9%
Lidocaine alone	1041	28.0%
Alpha_2_ + Ketamine	384	10.3%
Alpha_2_ + Ketamine + Butorphanol	147	4.0%
Alpha_2_ + Lidocaine	78	2.1%
Ketamine alone	70	1.9%
Alpha_2_ + Butorphanol	68	1.8%
Lidocaine + Ketamine	65	1.7%
Alpha_2_ + Lidocaine + Ketamine	47	1.3%
Alpha_2_ + Lidocaine + Butorphanol	18	0.5%
Other combinations	20	0.5%

Alpha_2_, Alpha_2_ agonists; CRI, continuous rate infusion.

**Table 11 animals-11-02549-t011:** Parenteral drugs administered for/during the immediate recovery period after general anaesthesia in 6701 horses.

Drugs and Combinations	*n*	%
Alpha_2_ alone	4220	63.0%
Alpha_2_ + Pure Opioids	197	2.9%
ACP + Alpha_2_	91	1.4%
Alpha_2_ + Partial/Agonists-Antagonists Opioids	65	1.0%
ACP alone	38	0.6%
Pure Opioid alone	37	0.6%
ACP + Alpha_2_ + Partial/Agonists-Antagonists Opioids	18	0.3%
Partial/Agonist-Antagonist Opioid alone	11	0.2%
ACP + Alpha_2_ + Pure Opioids	10	0.1%
ACP + Partial/Agonists-Antagonists Opioids	3	0.0%
Benzodiazepine alone	2	0.0%
ACP + Alpha_2_ + Pure Opioids *+* Partial/Agonists-Antagonists Opioids	1	0.0%
ACP + Pure Opioids	1	0.0%
Pure Opioids + Partial/Agonists-Antagonists Opioids	1	0.0%
None	2006	29.9%

ACP, acepromazine; Alpha_2_, Alpha_2_ agonists. Alpha_2_-agonists: 79 cases received more than one alpha_2_-agonist in the recovery.

**Table 12 animals-11-02549-t012:** Methods of induction of general anaesthesia for 6701 horses under inhalational anaesthesia only (INH), partial intravenous anaesthesia (PIVA), total intravenous anaesthesia (TIVA) and overall.

Induction	INH (*n* = 2282)	PIVA (*n* = 3718)	TIVA (*n* = 701)	Overall (*n* = 6701)
Personnel assisted	1498 (65.6%)	2010 (54.1%)	570 (81.3%)	4078 (60.8%)
Gate	353 (15.5%)	1207 (32.5%)	86 (12.3%)	1646 (24.6%)
Free	407 (17.8%)	479 (12.9%)	44 (6.3%)	930 (13.9%)
Table	18 (0.8%)	9 (0.2%)	0 (0%)	27 (0.4%)
Sling	6 (0.3%)	13 (0.3%)	1 (0.1%)	20 (0.3%)

**Table 13 animals-11-02549-t013:** Methods of recovery from general anaesthesia for 6461 horses after inhalational anaesthesia only (INH), partial intravenous anaesthesia (PIVA), total intravenous anaesthesia (TIVA) and overall.

	INH (*n* = 2225)	PIVA (*n* = 3543)	TIVA (*n* = 693)	Overall * (*n* = 6461)
Free	1049 (47.2%)	1770 (50.0%)	462 (66.7%)	3281 (50.8%)
Ropes	942 (42.3%)	1598 (45.1%)	116 (16.7%)	2656 (41.1%)
Manual	234 (10.5%)	175 (4.9%)	115 (16.6%)	524 (8.1%)

* 6701–6461 = 240 horses that were put to sleep (PTS) or died before recovery (i.e., premedication, induction or maintenance).

**Table 14 animals-11-02549-t014:** Drugs used for standing sedation in 1955 horses.

Drugs	PREM	MAIN—(Bolus)	MAIN-CRI	POST
Acepromazine	918	15	0	1
Xylazine	61	17	29	1
Detomidine	1574	1178	525	1
Romifidine	405	302	60	0
Medetomidine	11	8	10	0
Dexmedetomidine	4	7	4	0
Midazolam	15	61	34	0
Diazepam	3	10	1	0
Morphine	532	54	76	7
Methadone	223	20	4	2
Pethidine	2	0	0	0
Fentanyl	2	0	0	0
Buprenorphine	3	0	0	0
Butorphanol	922	437	94	0
Phenylbutazone	253	5	0	115
Flunixin	513	3	0	137
Meloxicam	39	2	0	14
Ketamine	0	8	24	0
Phenylephrine	0	1	3	14

CRI, continuous rate infusion; MAIN, maintenance; POST, postoperative period; PREM, premedication.

**Table 15 animals-11-02549-t015:** Drugs and different drug combinations used for premedication before standing sedations in 1955 horses.

Drugs and Combinations	*n*	%
Alpha_2_ + Partial/Agonists-Antagonists Opioids	588	30.1%
ACP + Alpha_2_ + Pure Opioids	393	20.1%
Alpha_2_ + Pure Opioids	338	17.3%
ACP + Alpha_2_ + Partial/Agonists-Antagonists Opioids	312	16.0%
ACP + Alpha_2_	203	10.4%
Alpha_2_ alone	93	4.8%
Alpha_2_ + Pure Opioids + Partial/Agonists-Antagonists Opioids	14	0.7%
ACP + Alpha_2_ + Pure Opioids + Partial/Agonists-Antagonists Opioids	10	0.5%
Pure Opioid alone	3	0.1%
Partial/Agonist-Antagonist Opioid alone	1	0.0%

ACP, acepromazine; Alpha_2_, Alpha_2_ agonists.

**Table 16 animals-11-02549-t016:** Drugs and different drug combinations used for continuous rate infusion in standing sedations in 1955 horses.

CRI Drugs	*n*	%
Alpha_2_ alone	448	22.9%
Alpha_2_ + Butorphanol	88	4.5%
Alpha_2_ + Morphine	66	3.4%
Alpha_2_ + Ketamine	10	0.5%
Alpha_2_ + Ketamine + Morphine	8	0.4%
Alpha_2_ + Ketamine + Butorphanol	4	0.2%
Alpha_2_ + Methadone	3	0.2%
Other combinations	6	0.3%
No CRI (only top-ups)	1322	67.6%

Alpha_2_, Alpha_2_ agonists; CRI, continuous rate infusion.

## Data Availability

Raw data, converted to a .csv file, are stored in the CEPEF4 metadata file. The most relevant results can be accessed at: https://cepef4.wordpress.com/preliminary-results/ (accessed on 6 June 2021).

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
