# Peer review of "Data Collection for the Fourth Multicentre Confidential Enquiry into Perioperative Equine Fatalities (CEPEF4) Study: New Technology and Preliminary Results"

_animals, 2021, doi:10.3390/ani11092549_

Round 1

Reviewer 1 Report

please see attached document

Author Response

Reviewer 1

-Comment: Thank you very much for giving me the opportunity to review this very interesting article, which describes the collection process and the presentation of preliminary data collected form questionnaire for inquiring peri-anaesthetic fatalities in horses. The authors should be congratulated for the initiation of this study, which was long awaited by the equine anaesthetists’ community. The results presented here confirm that the method is valid and practical, which justifies to continue the data collection process in the same way. Validating the method at this stage seems essential to me, as it guarantees that data collected throughout the whole study period will allow the evaluation of anaesthetic risk in horses and the identification of risk factors. It somehow guarantees that the time effort that will be spent by all the collaborating centres in collecting the data is not in vain. The final aim is the determination of the anaesthetic risk and specific risk factors in horses.

The article is well written. The introduction contains relevant information on the topic as well as the clearly formulated hypotheses. The description of the data and the results is very comprehensive. At this stage, there is little interpretation of the data, which is explained by the fact, that these preliminary data support the choice of the method and justify the continuation of data collection. I have two more general questions regarding the determination and the put to sleep data. It is not clear what made you decide that a horse that is euthanized during colic surgery is a colic put to sleep.

Some limitations are mentioned. Please find my specific comments below.

-Answer: Thank you very much for your kind words regarding our study and paper. Also, thanks for your comments which will improve the quality of the manuscript. Please see below the individual answers to each of your comments. We have also slightly modified some of the tables (i.e. Tables 2, 10, 13), but these changes are minimal and do not alter the meaning of the Results/Discussion.

-Comment: Lines 31-32: Consider rephrasing the introduction of the abstract to start with the ‘risk of peri- operative mortality in horses’ and not with ‘internet-based research’, as this is the main aim of the study.

-Answer: We have deleted the sentence “internet-based research” while maintaining the Background of the CEPEF2 study (lines 31 – 33). As suggested, we have included the “risk of peri- operative mortality in horses” in line 35. We have reworded some other parts of the abstract to comply with the word count (200 words).

-Comment: Line 33: I think only the data collection is internet based, while the cleaning process is not internet based (R). Consider rephrasing.

-Answer: This is correct. Lines 33 – 34 have been rewritten accordingly.

-Comment: Lines 35 - 36: “The objective of these descriptive preliminary results....” This sentence sounds a little bit like the objective of the preliminary results is to describe preliminary results. Maybe there is way to rephrase this sentence in a more concise way?

-Answer: “…of these descriptive preliminary results…” has been deleted.

-Comment: Lines 73-75: Introduction/Aims. I think it is worth to mention that at least in the overall CEPEF 4 study, the aim is to identify risk factors.

-Answer: This has been included in the last paragraph of the Introduction section.

-Comment: Line 88: I would delete ‘strictly’.

-Answer: “Strictly” deleted.

-Comment: Line 110: “... analgesia and analgesia.” I guess it is anaesthesia and analgesia.

-Answer: Correct. Amended.

-Comment: Line 112: consider rephrasing to “the level of training of the responsible anaesthetist”.

-Answer: This has been included.

-Comment: Line 125: I think the way the decision is take to grade a horse as PTS or as dead needs to be described more comprehensively. It is essential to exclude any bias coming from the intra- operative environment. Sometimes horses are euthanized for bad prognosis but also for financial aspects. I think only the minority is euthanized for lesions that can’t be repaired during surgery, but rather a mix of difficult surgical conditions, financial restraints, maybe pre-existing conditions (e.g. chronic lameness) but maybe also fatigue of the staff. I know that it will extremely difficult to differentiate all these factors. Also, once the horse is put to sleep it has no chance to die from complications anymore. So probably the worst cases, those with the highest chance of dying during the post-operative period are excluded. I can understand the argument, that dead is dead, but I would appreciate to learn more on how the horses were classified to either one category. Maybe the owners would like to comment on this in the discussion as well.

-Answer: "PTS" only gets allocated to cases that were PTS because of inoperable lesions, financial constraints etc. "Deaths" include horses that died or were euthanised because of complications incurrence once they had been anesthetised. This is included in section 2 of the Materials and Methods, and is defined in the same way as previous CEPEF studies.

-Comment: Line 130: What about financial reasons?

-Answer: This has been included in line 130.

-Comment: Line 185: consider replacing the second ‘available’ in this phrase.

-Answer: The second “available” has been replaced by “different”.

-Comment: Lines 205: Do the authors think that the final process of data cleaning, where the PI is talking to the centre ambassador could lead to more bias? Wouldn’t it be better to let the ambassador correct data that are simply highlighted in a table? Just a question. I don’t know the answer for this.

-Answer: We have included now in lines 205 – 207 that for every inconsistency indeed we sent the ambassador an Excel file for his/her centre. The final meeting was only to evaluate if all the information was correct, double-checking for missing data, final inconsistencies, etc. Therefore, we do not think that the meeting with the PI led to any kind of bias.

-Comment: Line 261: The figure seems a bit confusing. The scale gives the impression that there were quite a lot of cases in South America, although there were very few. The figure is slightly blurred in the pdf document I have received for review.

-Answer: In Figure 2 we have used a logarithmic scale (not linear) to emphasize those countries with higher number of cases.

-Comment: Line 273: You excluded 94 donkeys as you feel that the sample size was too small. You left field procedures in the data, although there were only 39. Is there a statistical reason for this? In figure 3, you mention the number of donkeys excluded, but not the number of ‘horses receiving no top- ups’. Maybe the number was n= 0. Then, please list the number for all excluded cases as n= 0. Like this, it will make sense 8752-96= 8656. If there were any horses excluded for the reasons ‘no top- up, not advanced imaging or surgery or out of the period’ then the numbers are not correct.

-Answer: We have excluded the donkeys as they are different species and the sample size was too small at this stage. However, field anaesthesia is a studied variable within those studies in horses.

With regard to “horses receiving no top-ups”, we have included now that n was indeed = 0. The same for advance imaging and surgery out of period. 

-Comment: Line 288: Table 2: ‘Colic Surgery’ as parameter with the variable ‘non colic surgery’. Although I do understand, I feel it is confusing. Consider rephrasing.

-Answer: Amended.

-Comment: Table 2: 4 horses were ventilated while understanding sedation. Was this CPAP or something like this. Just by curiosity.

-Answer: Thank you very much for this comment. Indeed, we have double-checked those cases and are mistakes. Those have been deleted and numbers corrected.

-Comment: Line 311: There are 35 non-colic death and 31 colic deaths. If the colic PTS had the ‘chance’ of being included in the ‘death’ category, there would be even a lot more colic death. The percentage of colic PTS is slighter higher than previously reported, so the question is, if these horses are just graded differently. Maybe something to discuss if confirmed in the final results.

-Answer: The classification of the deaths/PTS was done exactly in the same way as in previous CEPEF studies. This is stated in lines 124 – 125.

-Comment: Line 312 and elsewhere: Table 3: What let you make these categories? Why not dental or ophthalmology?

-Answer: Our classification was based on that of CEPEF2 (added diagnostic procedures). It is true that dental and ophthalmology cases have been included in the miscellaneous type. However, this is an interesting comment that will be taken into account in the future.

-Comment: Line 318 Table 4a: I can see, that there is nothing to do about cardiac arrest. It is a fair reason to die. But for horses dying of colic, they could have been prevented from doing so by surgery (I hope, at least). And the reason not to operate on them, is probably not only the chance of dying, which was probably not 100%, but based on many factors. This brings me back at my initial question. I think it would be good to mention this in the discussion.

-Answer: This is an interesting comment. However, in tables 4 a,b,c we are just focussing in noncolic deaths. As in CEPEF2, colic deaths were not investigated as they could be due to variable reasons. However, information about the colic deaths/PTS has been collected and will be studied in the future.

-Comment: Line 324: Table 4a and b: It is not always clear what is included in the “brief description of non colic death”. These seems to be a mixture of the pre-existing condition and the reason of death. Also the some words are not easy to understand, e.g. ‘down in van’ ... this means that the horse arrived lying in the horse box/trailer? So this would related to a pre-anaesthetic condition, while ‘unable to stand’ refers to the recovery? “Fixation exposed elbow fracture plate failed in recovery” not clear to me. Tibia fracture occurred during recovery? Not the ‘hit by car? I would suggest describing all these conditions a little bit more in detail as they are certainly very interesting for the reader. If space is an issue, I would reduce tables 3 a and b instead. Especially because at this point there is no analysis of number of death/PTS per condition.

-Answer: Thanks for your comment. Based on that we made some modifications in Table 4b. We think that for those noncolic deaths with ASA III, IV or V it would be interesting for the reader to know something about the pre-anaesthetic condition, therefore the we now entitle the first column in Table 4b as “Pre-existing condition and brief noncolic death description”. We have modified some of the cases and grouped those re-fractures. We hope this is clearer now.   

-Comment: Line 351: Table 6: Bolus - if parenteral”. Were there any non-parenteral treatments during maintenance? Maybe simply bolus IV or just bolus? Same in table 14.

-Answer: We included “if parenteral” to differentiate from those inhalants given during maintenance, which indeed cannot be administered by bolus. We have now included “if not inhalant”.

-Comment: Line 402: Table 14: Bolus if parenteral?

-Answer: Agreed. There are no inhalants. “If parenteral” has been deleted.

-Comment: Line 412: Table 16: Consider rephrasing ‘No CRI (top-ups)’ to ‘No CRI (only top-ups)’ or ‘only repeated boli’. It is a little bit confusing to read in the title drugs used for CRI and then to read in the table no CRI. This was also in some of the previous tables. In the table no combination with opiods is listed although in the text (line ) opiods as part of CRI protocols for standing sedation are mentioned.

-Answer: We had included “only” in Table 16 as suggested. Regarding to the second part of the comments, we have modified Table 16 and now give more details about the different combinations given as CRIs.

-Comment: Line 462: consider changing ‘future CEPEF4 study to ongoing CEPEF4 study.

-Answer: Amended to “current”.

-Comment: Line 521: Please change sentence to read: “...premedication with acepromazine alone has not been reported in the current study.”

-Answer: Amended.

-Comment: Line 656: I have not checked the format of the references.

-Answer: We have re-checked the references and the reference Dugdale & Taylor (2016) had been included twice by mistake (19 and 31). We have therefore deleted reference 31 and re-numbered the remaining accordingly.  

We have also re-checked the format of the references, those with “;” missing between the names of the authors have been corrected.

Reviewer 2 Report

Reviewer comments for manuscript ID animals-1307175 entitled ‘Data Collection for the Fourth Multicentre Confidential Enquiry of Perioperative Fatalities (CEPEF4) Study: New Technology and Preliminary Results

General Comments

A brilliant as well as unique attempt by the authors to collaboratively work on a very important and contemporary aspect of equine anaesthesia. Building upon the background of a previous study done 20 years ago is an excellent way to introspect and analyse the growth and development of equine anaesthesia in different clinical settings, case variety as well as geographically. The study design, data collection methodology and analyses have been clearly described and the results have been meticulously presented. COVID restrictions have lead to innovative study designing that should encourage peers to evolve research and data collection methods. Discussion is quite precise and relevant to the results presented. I congratulate the authors for their painstaking hard work that should be a benchmark for future sub studies on equine anaesthesia. I am very hopeful that this work has worldwide ramifications and should improve as well as encourage equine anaesthetists in the reduction of anaesthetic mortalities.

Specific Comments

Lines 17-18: Please reframe ‘to the lower rates seen in companion animals such as dogs and cats’ as ‘in comparison to small animal anaesthesia’

Line 21: Please rewrite ’6,701 general anaesthetics and 1,955 standing procedures’ as ‘6701 procedures under general anaesthesia and 1,955 procedures in standing sedation’

Lines 35-36: Please rewrite ‘6,701 general anaesthetics’ as ‘6701 procedures under general anaesthesia’

Lines 37-38: It seems that the authors mean to say the utility was 1%. Please clarify or rewrite clearly.

Lines 43-44: Incomplete sentence. Please complete it.

Line 68: Please replace ‘supposition’ with ‘hypothesis’.

Lines 129-30: For better comprehensibility of the reader in this sentence ‘Classification of outcomes was performed by XX and XX, and later confirmed by XX, XX and XX’ please place brackets ‘(see coding of data and anaesthetists under 6. Anonymity and confidentiality of each patient, owner and centre)’.

Line 288: Ref. Table 5: I think ‘Time of death’ is a better word than ‘Moment of death’

Line 374: Please clarify what bias did you anticipate with standing sedations?

Lines 375-78: Acute abdomen, time of presentation and other co-morbidities might have contributed to the mortality in standing sedations instead of anaesthetic/sedative agents. Please clarify.

Author Response

Reviewer 2

-Comment: A brilliant as well as unique attempt by the authors to collaboratively work on a very important and contemporary aspect of equine anaesthesia. Building upon the background of a previous study done 20 years ago is an excellent way to introspect and analyse the growth and development of equine anaesthesia in different clinical settings, case variety as well as geographically. The study design, data collection methodology and analyses have been clearly described and the results have been meticulously presented. COVID restrictions have led to innovative study designing that should encourage peers to evolve research and data collection methods. Discussion is quite precise and relevant to the results presented. I congratulate the authors for their painstaking hard work that should be a benchmark for future sub studies on equine anaesthesia. I am very hopeful that this work has worldwide ramifications and should improve as well as encourage equine anaesthetists in the reduction of anaesthetic mortalities.

-Answer: Dear Reviewer. Thank you very much for your kind words regarding our study/paper. Thank you also for your comments which have improved the quality of the manuscript. Please see below for the individual answers to each comment. We also have slightly modified some of the tables (i.e. Table 2, 10, 13); these changes are minimal and do not alter the meaning of the Discussion.

Specific Comments

-Comment: Line 19: Please reframe ‘to the lower rates seen in companion animals such as dogs and cats’ as ‘in comparison to small animal anaesthesia’.

-Answer: Amended to” in comparison with small animal anaesthesia”.

-Comment: Line 24: Please rewrite ’6,701 general anaesthetics and 1,955 standing procedures’ as ‘6701 procedures under general anaesthesia and 1,955 procedures in standing sedation’.

-Answer: This has been rewritten accordingly.

-Comment: Lines 35-36: Please rewrite ‘6,701 general anaesthetics’ as ‘6701 procedures under general anaesthesia’

-Answer: Amended.

-Comment: Line 40: It seems that the authors mean to say the utility was 1%. Please clarify or rewrite clearly.

-Answer: Good point. Rewritten.

-Comment: Lines 46-47: Incomplete sentence. Please complete it.

-Answer: “…is suitable…”. Is this the incomplete sentence? This sentence has been shortened due to the changes proposed by both reviewers and the limited wordcount (200 words).

-Comment: Line 71: Please replace ‘supposition’ with ‘hypothesis’.

-Answer: Amended.

-Comment: Lines 137-138: For better comprehensibility of the reader in this sentence ‘Classification of outcomes was performed by XX and XX, and later confirmed by XX, XX and XX’ please place brackets ‘(see coding of data and anaesthetists under 6. Anonymity and confidentiality of each patient, owner and centre)’.

-Answer: XX is for blinding, which refers to the initials of the authors. This is not related to section 6, which is about patients, owner or centre.

-Comment: Line 336: Ref. Table 5: I think ‘Time of death’ is a better word than ‘Moment of death’.

-Answer: Changed accordingly. This has also been clarified in the text in Materials and Methods, line 125.

-Comment: Line 433 – 434: Please clarify what bias did you anticipate with standing sedations?

-Answer: This has been clarified in the standing sedation section, lines 587 – 588.

-Comment: Lines 504: Acute abdomen, time of presentation and other co-morbidities might have contributed to the mortality in standing sedations instead of anaesthetic/sedative agents. Please clarify.

-Answer: Yes, agreed. We have now included not the ASA classifications of the standing sedations in lines 504 – 505, which were not included as for the general anaesthetics. Hopefully this is clearer now.